TOPICAL REVIEW

# Glial cells in the heart: Implications for their roles in health and disease

Svetlana Mastitskaya[1] , Rimma Dugarova[1] and Shefeeq M. Theparambil[2]

[1] Translational Health Sciences Department, Bristol Medical School, University of Bristol, Bristol Royal Infirmary, Upper Maudlin Street, Bristol, BS2 8HW, UK
[2] Biomedical and Life Sciences Department, Lancaster University, Lancaster, UK

Handling Editors: Bjorn Knollmann & T. Alexander Quinn

The peer review history is available in the Supporting Information section of this article (https://doi.org/10.1113/JP286598#support-information-section).

**Abstract figure legend** Schematic representation of cardiac autonomic ganglia within epicardial fat pads (posterior heart surface shown), containing vagal postganglionic neuron cell bodies, associated fibres, and glia. These ganglia receive cholinergic input from vagal preganglionic neurons and adrenergic input from sympathetic postganglionic neurons, forming part of the intrinsic cardiac autonomic network that regulates heart rate, rhythm, conduction and contractility. Top inset: cardiac tripartite synapse showing autonomic input (acetylcholine (ACh)/noradrenaline (NA)) to a pacemaker (P) cell and a satellite glial cell in the sinoatrial node. Satellite glia form synapse-like contacts with P cells, modulate neurons and P cells via purinergic and glutamatergic signalling (replenishing neuronal glutamine via the glutamate–glutamine cycle), and release nerve growth factor (NGF) and other neurotrophic factors that support neural and cardiovascular function. Bottom inset: neurons are ensheathed by satellite glia; some associate with Schwann cells. Additional nonneuronal cells (e.g. fibroblasts, myoblasts) support plexus structure and function.

**Svetlana Mastitskaya** is a British Heart Foundation Intermediate Basic Science Research Fellow and Senior Lecturer at the University of Bristol. She holds a PhD in cardiac regeneration and specialises in Cardiovascular Neuroscience. Her research focuses on how neural and humoral mechanisms, particularly vagus nerve signalling, regulate coronary blood flow in health and disease, and how these brain–heart communication pathways can be therapeutically targeted to improve cardiovascular health.

The Journal of Physiology

**Abstract** Glial cells are essential regulators of brain homeostasis by orchestrating neuronal function, metabolism and immune responses. However, much less is known about peripheral glial cells, particularly those in the heart. This review explores the development, types and functions of cardiac glial cells, including Schwann cells, satellite glial cells and recently identified cardiac nexus glia, with some reference to their central nervous system counterparts. The heart's autonomic nervous system consists of sympathetic and parasympathetic nerve fibres, primarily located in the epicardial fat pads within the transverse and oblique sinuses and around the roots of the great vessels. Schwann cells support cardiac repair by myelinating neurons and modulating inflammation, while satellite glial cells regulate the neuronal microenvironment, influencing heart rate and rhythm. Cardiac nexus glial cells interact with both sympathetic and parasympathetic pathways to regulate heart function. Understanding the roles of cardiac glial cells could provide new insights into neuro-cardiac interactions and reveal potential therapeutic targets for cardiac disorders.

(Received 25 February 2025; accepted after revision 15 August 2025; first published online 6 September 2025)

**Corresponding author** S. Mastitskaya: Translational Health Sciences Department, Bristol Medical School, University of Bristol, Bristol, UK. Email: svetlana.mastitskaya@bristol.ac.uk

## Introduction

For the past 30 years, significant advancements have been made in the field of glial biology, revealing the intricate roles for glial cells in the nervous system beyond their traditionally perceived supportive functions. Glial cells, also known as neuroglia, are non-neuronal cells that do not generate action potentials but play essential roles in neural support and maintenance. Glial cells in the central nervous system (CNS) include astrocytes, which maintain homeostasis and contribute to the blood–brain barrier; oligodendrocytes, which generate myelin for axonal insulation; and microglia, which function as resident immune cells (Azevedo et al., 2009). Not only are glial cells integral to brain protection, neural development and functional regulation (Bear & Caspary, 2024; Oikonomou & Shaham, 2011; Rasband, 2016; Shaham, 2015), but their dysfunctions are also linked to neurodegenerative diseases such as amyotrophic lateral sclerosis and Alzheimer's disease (Hashioka et al., 2021; Rasband, 2016). The high heterogeneity of cells grouped under the single term 'glia' is revealed through gene profiling of human and mouse glial cells, uncovering distinct genetic profiles even within the subtypes of microglia, astrocytes and oligodendrocytes (Hickman et al., 2013; Masgrau et al., 2017; Zhang et al., 2016).

While glial research has predominantly focused on the CNS, where the diversity and critical functions of glial cells have been well-documented, the role of glial cells in the peripheral nervous system (PNS) remains less explored. This leaves a gap in understanding the comprehensive roles glial cells play in peripheral tissues, despite their crucial involvement in homeostasis, development and disease processes.

Glial cells in the PNS, much like their CNS counterparts, provide essential support to neurons while also exhibiting unique properties and functions tailored to peripheral tissues. The PNS serves as a conduit for communication between the CNS and the rest of the body, encompassing sensory and motor neurons along with various supporting glial cell types. Two primary glial cell types in the PNS are Schwann cells and satellite glial cells. Schwann cells myelinate peripheral axons, ensuring rapid signal conduction and aiding in nerve regeneration following injury, while satellite glial cells regulate the microenvironment of neuronal cell bodies within ganglia, providing structural and nutritional support (Jessen & Mirsky, 2005).

Peripheral glial cells are highly specialised, with adaptations for diverse roles in different tissues. For instance, at the neuromuscular junction, glial cells facilitate synaptic transmission and contribute to synaptic plasticity, underscoring their active involvement in motor control and peripheral synaptic function (Ko & Robitaille, 2015; Perez-Gonzalez et al., 2022). Similarly, in the enteric nervous system (ENS), glial cells outnumber enteric neurons and are integral to gastrointestinal function, coordinating with neurons to regulate motility, secretion, maintenance of the integrity of the mucosal barrier, and responses to injury or inflammation through antigen presentation and secretion of cytokines (Rühl, 2005). Their dysfunction has been linked to disorders such as irritable bowel syndrome and inflammatory bowel disease, demonstrating the significance of glial function beyond the CNS (Bear & Caspary, 2024).

Recent studies have also begun to shed light on the unique population of glial cells associated with the heart, referred to as cardiac glia. These cells,

particularly the newly identified cardiac nexus glia (CNG), have been implicated in the modulation of cardiac function (Gunsch et al., 2023). The heart's autonomic regulation is mediated by the interplay between the sympathetic and parasympathetic nervous systems, which influences heart rate, contractility and vascular tone. Sympathetic neurons, located in paravertebral ganglia, stimulate the sinoatrial node (SAN) via noradrenaline release, driving the acute physiological stress response. Conversely, parasympathetic neurons, situated within cardiac ganglia, promote the homeostatic adaptations, rest and regeneration state through acetylcholine release (Kandel, 2013). While much is known about neuronal contributions to these processes, the role of cardiac glia in modulating autonomic control and maintaining cardiac homeostasis is an emerging area of investigation. Like their CNS and ENS counterparts, cardiac glial cells may participate in neuroimmune interactions and contribute to homeostasis. However, their precise functions, cellular diversity and interactions with cardiac neurons remain poorly understood. Investigating cardiac glia could offer valuable insights into the autonomic regulation of the heart and unveil potential therapeutic targets for cardiovascular disease.

## Development of glial cells in the peripheral nervous system

The development of glial cells differs significantly between the CNS and the PNS. In the CNS, glial cells originate from the neural tube, which gives rise to the brain and spinal cord (Kessaris et al., 2008). Meanwhile, glial cells in the PNS emerge from the neural crest, a group of cells that migrate to various parts of the body during embryonic development (Donoghue et al., 2008). This distinction is important as it underscores the diverse origins and developmental pathways that contribute to the specialization and function of glial cells within different areas of the nervous system.

In the process of neurulation, the dorsal neural plate of the embryo folds inward to form the neural groove, which eventually closes to create the neural tube. Neural crest cells originate from the tips of these folds. This population of cells then separates into two main streams: lateral migration leads to the formation of melanocytes in the skin, while dorsal migration gives rise to neurons in dorsal root ganglia (DRG) and various types of glial cells, as well as autonomic neurons and chromaffin cells (Jessen & Mirsky, 2005).

The Schwann cell progression includes three temporary cell groups. After neural crest cells, they proliferate into Schwann cell precursors (SCPs), whose phenotypic profile is characterised by expression of calcium-dependent cell–cell adhesion glycoprotein Cad19 (Takahashi & Osumi, 2005), brain fatty acid-binding protein (BFABP), protein zero (P0), and desert hedgehog, all of which are also exclusive for this profile of cells. The second group after SCPs consists of immature Schwann cells (SCs), which eventually differentiate into mature myelinating and non-myelinating (Remak) SCs in the PNS. The markers of the late SC development are glial fibrillary acidic protein (GFAP) and S100 calcium-binding protein B (S100B) (Jessen et al., 2015). S100B is a protein commonly found in glial cells, while GFAP is an intermediate filament protein predominantly expressed in astrocytes and some other types of glial cells. In addition, S100B is a part of the huge S100 protein group, and the levels of this biomarker increase in brain and heart injury, especially after ablation procedure for atrial fibrillation (Bychkov et al., 2022; Scherschel et al., 2019).

The development of satellite glial cells (SGCs) involves several distinct stages similar to the differentiation of SCs. SGCs within DRG differentiate under the influence of various mediators such as neuregulin and Notch ligands. Following this, SGCs undergo a maturation stage where they express factors specific to them, such as S100B, GFAP and various purinergic receptors, including P2Y12 and P2X receptors. Purinergic P2Y12 receptors are involved in sensing extracellular nucleotides and are often used as markers for SGCs due to their role in glial–neuronal communication and function within sensory ganglia (Hanani & Spray, 2020).

Figure 1 and Table 1 summarise the signature characteristics of various glial populations across autonomic ganglia.

**Schwann cells and their functions, notably in the heart.** SCPs support the survival of developing neurons (Jessen et al., 2015). Studies have demonstrated this by removing glial cells during the embryonic development of mice. Mouse embryos lacking the ErbB2 or ErbB3 neuregulin receptors experienced the death of motor and sensory limb neurons on embryonic days 14 and 18, respectively, although these cells were initially generated in normal numbers (Riethmacher et al., 1997). Furthermore, published results on heterozygous SOX10 mice (carrying one functional and one non-functional copy of SOX10 gene, a key transcription factor in glial development) (Britsch et al., 2001) and the lack of neuregulin-1 (Garratt et al., 2000) signalling have demonstrated similar outcomes. These mice exhibited severe deficits in glial cell development, leading to widespread neuronal death and significant impairments in peripheral nerve function. This evidence underscores the critical importance of SCPs and their associated signalling pathways in the maintenance and survival of neurons during embryonic development.

Myelinating SCs create the myelin sheath surrounding axons, which helps in electrical signal transmission. Remak SCs are also critical for optimal PNS function and

ensheathe smaller axons, keeping each distinct from each other to form Remak bundles. It is these Remak SCs which are crucial for repair and regeneration after an injury within the PNS. Mature states of SCs as either myelinating or Remak are not fixed, however, and following nerve injury, SCs can dedifferentiate and proliferate to support axonal regrowth and recovery (Jessen et al., 2015), as demonstrated in two forms of severe peripheral nerve injury: axonotmesis and neurotmesis. Firstly, in axonotmesis the nerve is a crushed or stretched so the axon itself is damaged, but the surrounding sheath and the Schwann cells remain intact. Conversely, neurotmesis describes complete division of the nerve and surrounding sheaths. Nerve recovery and/or regeneration between these injuries varies, with regeneration possible rapidly, in a matter of weeks following axonotmesis, *versus* slow and incomplete regeneration following neuro-

tmesis, with limited functional recovery, even though the distance between two ends looks like a microscopic gap where cells create a cellular bridge (Höke & Brushart, 2010). Wallerian degeneration follows both types of traumatic peripheral nerve injury, orchestrated by immune and non-immune cells including SCs. This involves anterograde degeneration of the distal end of the axon from the point of injury towards the periphery (Gaudet et al., 2011), including breakdown of the axon and myelin sheath distal to the injury site. Many cell types and molecules are activated and released during Wallerian degeneration; however, as axons can regenerate following axonotmesis but not neurotmesis, the cellular and molecular events of Wallerian degeneration may be disparate across these injuries (Rotshenker, 2011).

Given these fundamental roles of SCs, their function in the heart, particularly in repairing heart damage,

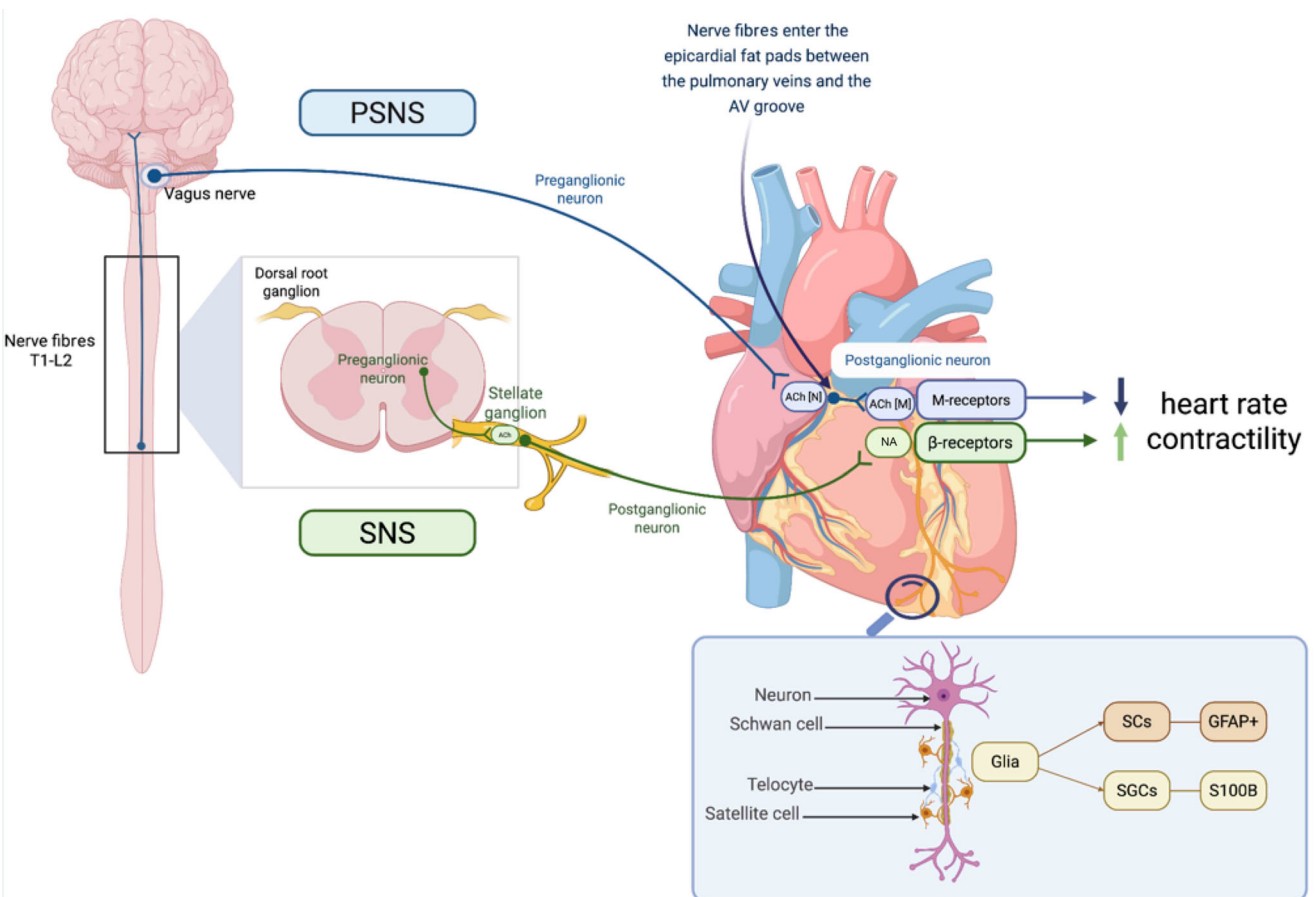

**Figure 1. The cardiac autonomic nervous system with glial cells and their markers**
Cardiac autonomic nervous system (ANS) comprises the sympathetic (SNS) and parasympathetic (PSNS) nervous systems. The SNS preganglionic axons originate from the thoracic spinal cord (T1–L2), releasing acetylcholine (ACh), which activates postganglionic neurons. These neurons release noradrenaline (NA), acting on $\beta$-adrenergic receptors on the heart, increasing heart rate and contractility. The PSNS primarily involves the vagus nerve. Its preganglionic axons release ACh to activate postganglionic neurons, which then release ACh acting on muscarinic receptors on the heart. Glial cells: Schwann cells (SCs): identified by S100 calcium-binding protein B (S100B), they myelinate peripheral axons for rapid signal conduction; satellite glial cells (SGCs): found in ganglia, they support neurons and are identified by glial fibrillary acidic protein (GFAP). Created with BioRender.

**Table 1. Characteristics of glial cell populations across cardiac autonomic ganglia**

| Glial population | Location | Markers | Functions |
|---|---|---|---|
| Satellite glial cells | DRG, nodose ganglia, stellate ganglia, intrinsic cardiac ganglia | GFAP, S100B, Kir4.1, P2X7, P2Y12, FABP7 | Neurotransmitter buffering, K$^+$/Ca$^{2+}$ homeostasis, cytokine release, neuroinflammation (Hanani & Spray, 2020; Jessen & Mirsky, 2005; Mapps et al., 2022) |
| Schwann cells | Peripheral nerves, intrinsic cardiac ganglia | S100B, GFAP, Ncmap | Axonal insulation, support and regeneration, neurotransmitter modulation (Jessen et al., 2015; Mapps et al., 2022) |
| Cardiac nexus glia | Epicardial ganglia and cardiac conduction system | GFAP, S100B, SOX10 | Synaptic modulation in cardiac ganglia (Kikel-Coury et al., 2021) |

DRG, dorsal root ganglia; FABP7, brain-type fatty acid-binding protein; GFAP, glial fibrillary acidic protein; Ncmap, non-compact myelin-associated protein; S100B, S100 calcium-binding protein B.

is significant. SC loss in myocardial infarction (MI) contributes to sympatho-vagal imbalance and the development of post-MI arrhythmias (Zhang et al., 2010), and transplantation of SCs is suggested as one of the tissue regeneration strategies post-MI. The healing process after MI includes Schwann cell proliferation and axonal regrowth in association with scar tissue formation (Vracko et al., 1991). In response to cardiac injury, SCs can exhibit plasticity, similar to their response in peripheral nerve injury, aiding in the repair and regeneration of damaged cardiac tissue by synthesis and release of neurotrophic factors promoting angiogenesis, cell survival and tissue repair (Jessen & Mirsky, 2019; Zhang et al., 2010).

Moreover, SCs have been implicated in modulating inflammatory responses in the heart. Their interaction with immune cells and secretion of neurotrophic factors can influence cardiac healing processes. For instance, SC-derived neuregulin-1 has been shown to promote cardiac repair and improve outcomes following myocardial infarction by enhancing cardiomyocyte survival and function (Liu et al., 2006). This highlights the potential therapeutic applications of targeting SCs in cardiac repair strategies.

**Satellite glial cells and their functions, notably in the heart.** SGCs are specialized peripheral glia found exclusively in the PNS where they ensheath the cell bodies of neurons in sensory, parasympathetic and sympathetic ganglia (Hanani & Spray, 2020; Pannese, 2010; Qarot et al., 2024). Electron microscopy has shown a consistently narrow extracellular cleft (∼15–20 nm) between neuronal membranes and SGC sheaths, forming discrete 'neuron–glia units' (Pannese, 2010; Qarot et al., 2024). This intimate anatomical arrangement facilitates functional coupling via ion channels (e.g. Kir4.1, P2X7, P2Y12) and paracrine mediators such as ATP, cytokines and nitric oxide, demonstrated by electro-

physiology, calcium imaging and tracer-dye studies (Andreeva et al., 2022; Chen et al., 2022; Enes et al., 2020; Feldman-Goriachnik & Hanani, 2021; Hanani & Spray, 2020; Huang et al., 2013). Following nerve injury, SGCs show hallmark activation characterized by GFAP upregulation, enhanced gap-junction coupling and increased pro-inflammatory signalling, all of which modulate neuronal excitability and contribute to chronic pain (Bang et al., 2024; Qiao, 2024; Zhou et al., 2019).

SGCs in cardiac autonomic ganglia provide structural support, protect neurons and regulate the perineuronal microenvironment (similar to the astrocytes in the CNS); they are also involved in the regulation of heart rate and rhythm, though underlying mechanisms remain incompletely understood (Andreeva et al., 2022; Hanani & Spray, 2020). Within cardiac ganglia, SGCs form functional networks around neurons and respond to microenvironmental changes, including inflammation or injury, by releasing signalling molecules such as ATP, glutamate and cytokines. This bidirectional signalling modulates autonomic control of cardiac function, facilitating physiological adaptation. Disrupted SGC–neuron communication during inflammation has been linked to cardiac dysregulation and arrhythmogenic pathologies (Ajijola et al., 2017; Xie et al., 2017).

SGCs in sympathetic ganglia contribute to neurotransmitter regulation and ion homeostasis through multiple mechanisms. They are involved in the uptake and release of neurotransmitters such as ATP and glutamate, facilitated by their expression of purinergic receptors (P2X and P2Y) and glutamate transporters (Kushnir et al., 2011; Vit et al., 2008; Weick et al., 2003). They prevent excitoxicity and maintain neuronal homeostasis by clearing excess neurotransmitters from the extracellular space. Additionally, SGCs maintain ion balance through ion channels and transporters that regulate extracellular concentrations of potassium (K$^+$) and calcium (Ca$^{2+}$) (Vit et al., 2008). Potassium channels on SGCs uptake excess

$K^+$ ions released during neuronal activity, preventing hyperkalaemia that could disrupt cardiac action potential propagation and lead to arrhythmias. Similarly, SGCs express calcium-binding proteins and calcium pumps to regulate $Ca^{2+}$ levels. By forming tight junctions with neurons, SGCs establish a controlled microenvironment critical for precise neurotransmitter and ion regulation, indirectly influencing cardiac function via modulation of sympathetic outflow to the heart (Hanani & Spray, 2020).

**Cardiac nexus glia.** CNG, also known as cardiac glial cells, are recently described GFAP-positive glial cells residing in the outflow tract of the heart (Kikel-Coury et al., 2021). These cells, conserved across zebrafish, mice and humans, originate from the hindbrain neural crest, migrate to the heart and use Meteorin signalling via Jak/STAT3 for differentiation. In the adult heart, CNG form a net-like morphology, localising with neurons and interacting with synapses, and are integral part of the cardiac autonomic nervous system (ANS).

CNG facilitate communication between the CNS and cardiac tissue, support nerve regeneration, and are involved in regulation of heart rate and cardiac function through their interactions with both sympathetic and parasympathetic nerves (Fig. 1) (Jessen et al., 2015).

The ANS is composed of neurons and glial cells, which are embedded in the $HCN4^+$ cell meshwork forming the sinoatrial node. Here, as in the brain, the distinct cell clusters oversee different functions, each specialised for action potential firing rates. Cells of parasympathetic and sympathetic nervous systems in the SAN create autonomic plexi and modulate heart rate and rhythm. However, a novel model has been proposed suggesting that the SAN presents itself as a heterogeneous combination of structured populations of the cells: autonomic neuron plexi from parasympathetic and sympathetic nervous systems, peripheral $GFAP^+/S100B^+$ glial cells, telocytes and newly identified $S100B^+/GFAP^-$ cells in combinations with pacemaker cells (Bychkov et al., 2022).

A recent study has added new insight to the model of heart rate regulation, in proposing that parasympathetic stimulation not only decreases the action potential (AP) rate of pacemaker cells but also suppresses the activity of modules that generate higher AP rates, thereby unmasking modules that operate at lower rates. Therefore, the SAN uses distinct cell clusters for precise control of heart rate, with the PSNS acting as a powerful downshifter, over-riding $\beta$-adrenergic receptor stimulation and exhibiting stronger effects in its presence. This system balances robustness and flexibility, while parasympathetic-induced dormancy of certain SAN cells conserves energy by reducing AP firing (Maltsev et al., 2023).

To investigate the mechanisms of CNG heart rhythm regulation via the ANS, Kikel-Coury et al. (2021) employed two zebrafish models examining the CNG absence: a *metrn−/−* mutant and laser ablation of $nucGFP^+$ cells. The *metrn−/−* mutants are zebrafish that have a deletion or inactivation of the meteorin gene, which is involved in glial cell development. The $nucGFP^+$ cells are cells that have been genetically modified to express green fluorescent protein specifically in their nuclei. This enables identification and selective ablation of these cells for functional investigation. Treatment with the non-selective $\beta$-adrenergic agonist isoproterenol increased heart rate in control zebrafish, but this response was abolished in *metrn−/−* and ablated animals, indicating a crucial role of CNG in the sympathetic response. Similarly, the parasympathetic agonist carbachol reduced heart rate in control zebrafish, and this effect was again negated in CNG-deficient models. Ablation of the right ventricular outflow tract (a portion of either the left ventricle or right ventricle of the heart which passes blood from the ventricular chamber to the aorta and pulmonary artery, respectively) $nucGFP^+$ cells demonstrated location-dependent regulation by CNGs, with significant effects on heart rate modulation. Furthermore, increased sympathetic activity in *metrn−/−* animals induced severe ventricular fibrillation, supporting the role of CNGs in regulating cardiac rhythm through ANS modulation (Kikel-Coury et al., 2021).

## Developmental roles of glial cells in the heart

**Reinnervation of the heart after heart transplantation and role of glial cells.** The human heart possesses an intrinsic cardiac neural network (ICN) capable of autonomously regulating rhythm following heart transplantation (HTx). However, the transplantation completely severs extrinsic autonomic inputs to the donor heart. This denervation results in impaired autonomic regulation of heart rate, contractility and vascular tone immediately post-surgery. Reinnervation – particularly sympathetic reinnervation – can occur to some degree post-transplant, although the timing and functional impact of both sympathetic and parasympathetic reinnervation are still not fully understood and remain areas of active investigation.

The first signs of sympathetic reinnervation of the post-transplanted heart were reported to occur 6 months after surgery, and the reinnervation of the SAN occurs within 18 months, as evidenced from heart rate variability (HRV) analysis (Christensen et al., 2021, 2022; Weiner et al., 2025) and positron emission tomography studies (Bengel et al., 2002). Reinnervation vitally improves quality of life by enhancing exercise resistance (Velleca et al., 2023) and is associated with significantly improved survival (median survival time 19.9 years in reinnervated patients compared to 14.4 years in non-reinnervated patients) (Weiner et al., 2025).

Meanwhile, parasympathetic reinnervation of vagal nerve fibres is slower and less common (Christensen et al., 2022). Some researchers have observed high frequency HRV indicating parasympathetic activity already 6-12 months after HTx (Imamura et al., 2014), others report no signs of PSNS regrowth up to 10 years after HTx (Beckers et al., 2004; Lee et al., 2016).

While specific studies on the role of glial cells in cardiac reinnervation post-HTx are limited, insights from peripheral nerve repair suggest their critical supportive role. SCs and SGCs secrete neurotrophic factors such as nerve growth factor (NGF), brain-derived neurotrophic factor (BDNF) (Elia et al., 2021) and glial cell line-derived neurotrophic factor (GDNF) (Stanga et al., 2021). For instance, the secretion of NGF, BDNF and GDNF in co-culture systems demonstrated the supportive role of glial cells in nerve regeneration (Feng et al., 2012). These neurotrophic factors facilitate axonal growth and synaptic plasticity by binding to specific receptors on the surface of neurons. NGF binds to the receptor tropomyosin receptor kinase A, promoting neuronal survival and axon outgrowth. The abundant presence of NGF$^+$ glial cells in the SAN and atrioventricular node (AVN) (Kanemaru et al., 2023) suggests the crucial role of these cells in facilitating reinnervation of the nodal tissue post-HTx. BDNF binds to the receptor tropomyosin receptor kinase B, enhancing the growth and differentiation of new neurons and synapses (Bathina & Das, 2015). GDNF, through the GFR$\alpha$1 receptor and RET tyrosine kinase, supports the survival and maintenance of dopaminergic and motor neurons. In the PNS, GDNF promotes muscle trophism, presynaptic maturation and acts as a neuromodulator of synaptic transmission (Stanga et al., 2021). These interactions not only promote the structural integrity of neurons but also enhance their functional connectivity, which is critical for effective reinnervation (Palasz et al., 2023).

Emerging research also highlights the role of metabolic support in nerve regeneration. Sundaram et al. (2023) have demonstrated the crucial role of leptin in Schwann cells' mitochondrial function during nerve repair. The interaction between adipocytes and SCs supports the breakdown and resynthesis of myelin, thus meeting the high metabolic demands of nerve repair. This metabolic cooperation between adipocytes and SCs, which accelerates peripheral nerve repair, could be particularly important in the reinnervation of autonomic cardiac ganglia, which are embedded in the epicardial fat pads (Coote, 2013). In addition, glial cells modulate the inflammatory response by releasing cytokines that recruit and activate immune cells, which are essential for clearing debris and creating a favourable environment for nerve regeneration (Hanani & Spray, 2020). They also provide metabolic support to regenerating neurons by maintaining ion homeostasis and supplying essential nutrients (Palasz et al., 2023).

No direct studies have yet examined the role of cardiac glia such as nexus glia, SGCs or SCs in cardiac reinnervation post-transplantation. However, their known functions in aiding nerve regeneration, establishing functional synapses and integration into existing neural circuits suggest they may influence reinnervation pathways and support regenerating axons. This possibility remains hypothetical and warrants further investigation.

**Role of cardiac glia in key signalling pathways that guide heart development.** The development of the heart involves the precise regulation of several signalling pathways. These pathways (including Wnt/$\beta$-catenin, bone morphogenetic protein (BMP), Notch, sonic hedgehog, fibroblast growth factor (FGF), transforming growth factor-beta (TGF-$\beta$), and phosphoinositide 3-kinase/AKT) are integral to both cardiac development and the formation and function of glial cells in the heart.

Understanding how these signalling pathways influence glial cell development can provide insights into potential therapeutic targets for cardiac and neurological disorders. For instance, promoting Wnt signalling might aid in the proliferation of cardiac glia, which could support neuronal survival and function in the heart after cardiac injuries. Secondly, by targeting BMP signalling, it may be possible to enhance the differentiation of glial cells that support cardiac neurons to prevent congenital heart defects. Thirdly, promoting Notch signalling, crucial for the development of the cardiac conduction system, could be useful in treating arrhythmias. Furthermore, enhancing Hedgehog and FGF signalling can improve the proliferation and function of cardiac glial cells, which could be particularly beneficial in promoting cardiac repair after myocardial infarction by supporting creating new communication between cardiomyocytes.

Table 2 summarizes the main signalling pathways which guide the glial development in the heart, detailing their roles in heart development and mechanisms of action.

### Functional roles of glial cells in cardiac physiology

The supportive and metabolic roles of glia around neurons in the CNS have been well explored (Allen & Lyons, 2018; DeSantis & Smith, 2021; Hanslik et al., 2021; Rasband, 2016), and the association of glial dysfunction with neurodegenerative diseases such as Alzheimer's and Parkinson's diseases are being actively studied (Elia & Fossati, 2023; Hanslik et al., 2021; Hashioka et al., 2021). Ageing leads to increased pro-inflammatory cytokines, mitochondrial activity and lipid accumulation in glial cells, causing

**Table 2. Key signalling pathways that guide cardiac glial development**

| Signalling pathway | Role in heart development | Mechanism of action via cardiac glia development |
|---|---|---|
| Wnt signalling | Regulates cell proliferation, differentiation and migration during early heart development | Stabilization of $\beta$-catenin, interaction with transcription factors to influence gene expression required for glial differentiation and maturation (Horitani & Shiojima, 2024) |
| BMP signalling | Formation of cardiac mesoderm and heart structures | Activation of SMAD proteins that regulate gene expression and promote differentiation of neural crest cells into glial cells (Garside et al., 2013; Zheng et al., 2021) |
| Notch signalling | Plays roles in cardiac cell differentiation, morphogenesis and the formation of the conduction system | Notch receptors interact with ligands and release NICD, which affects gene transcription of glia (Garside et al., 2013; Luxán et al., 2016) |
| Hedgehog signalling | Develops outflow tract and ventricular myocardium | Sonic hedgehog ligand binds to receptor and activate GLI transcription factors which responsible to the glia proliferation and differentiation (Goddeeris et al., 2007; Kong et al., 2020) |
| FGF signalling | Promotes proliferation and differentiation of cardiac progenitor cells | FGF ligands bind to receptors and trigger MAPK/ERK signalling cascades for proliferation and differentiation glial cells (Itoh et al., 2016) |
| TGF-$\beta$ signalling | Regulates extracellular matrix production and epithelial-mesenchymal transition | TGF-$\beta$ ligands bind to receptors activate SMAD proteins that control gene expression necessary for glia differentiation and extracellular production (Garside et al., 2013; Sridurongrit et al., 2008) |
| PI3K/AKT signalling | Promotes cell survival and growth during heart development | Activation of AKT kinase regulate targets related to cell survival and metabolism (Jin et al., 2020) |

BMP, bone morphogenetic protein; ERK, extracellular signal-regulated kinase; FGF, fibroblast growth factor; MAPK, mitogen-activated protein kinase; NICD, Notch intracellular domain; PI3K, phosphoinositide 3-kinase; TGF-$\beta$, transforming growth factor-beta.

oxidative stress. Additionally, glial cells' ability to respond to injury and clear glutamate diminishes, leading to neuronal atrophy and synapse regulation dysfunction, which underlie neurodegeneration.

Conversely, the role of glia in the PNS remains less investigated (Gunsch et al., 2023). A review by Gunsch et al. (2023) looked to summarize the modern consensuses about functions of glia in the PNS. Their key messages include description of glial functions in different tissues such as nerve bundles, gastrointestinal, muscular, splenic, lymphatic, white and brown adipose tissue. However, they devote only a couple of paragraphs to cardiac glia, referencing just a single study by Kikel-Coury et al. (2021).

**Insights from transcriptomics studies.** Recent advances in single-cell and spatial transcriptomics have expanded our understanding of peripheral glial cell heterogeneity and function. Single-cell transcriptome profiling of sympathetic (superior cervical ganglion) and sensory (DRG) ganglia revealed five types of satellite glial cells (Mapps et al., 2022). One subtype was specific to sympathetic ganglia, one to sensory ganglia, and three had overlapping profiles: immune response SGCs, general resident SGCs characterised by the expression of cell

adhesion markers, and immediate early gene-expressing SGCs. Moreover, the identified subtypes of satellite glia can be distinguished from Schwann cells and astrocytes by several uniquely expressed genes: *Fabp7* (fatty acid binding protein 7), high levels of genes associated with fatty acid synthesis (including *ApoE*) and genes involved in mitochondrial $\beta$-oxidation (*Acaa2*, *Acadl*, *Acadm*, *Acsbg1* and *Eci1*). In contrast, Schwann cells are enriched for genes involved in myelination (*Ncmap*, non-compact myelin-associated protein) and genes associated with sphingolipid synthesis (*Fa2h*, *Samd8*, *Sptlc2* and *Ugt8a*). A differential expression profile for adhesion protein between Schwann cells and SGCs was also demonstrated (Mapps et al., 2022). Additionally, SGCs in sensory ganglia were enriched in Connexin 43, endothelin receptor B, MLC1 (modulator of VRAC current 1) and transcripts associated with cholesterol biosynthesis and turnover (Mapps et al., 2022).

A spatial transcriptomics study of human heart cardiac niches by Kanemaru et al. (2023) used the pan-glial markers *PLP1*, *NRXN1* and *NRXN3* to identify glial cells. They discovered a rich population of glial cells expressing *NGF* in the SAN and AVN, mostly localised to the central region of the nodal tissue. These glial

cells formed synapse-like connections with pacemaker P cells (pacemaker cells) through neurexins (NRXN1, NRXN3), glutamate and angiotensin II (AngII) signalling, supporting glutamatergic signalling (by replenishing neuronal glutamine pool via the glutamate–glutamine cycle) and release NGF to interact with autonomic neurons. Additionally, NGF$^+$ glial cells in SAN were shown to interact with fibroblasts. Overall, NGF$^+$ glia in SAN was shown to have an astrocyte-like supportive role. Furthermore, these NGF$^+$ glial cells were identified as the most abundant source of NGF, to which the receptors NGFR and NTRK1 are expressed in the right atrial ganglionated plexi, promoting autonomic innervation of the nodal tissue and supporting cardiac conduction system (Kanemaru et al., 2023).

Another important single-cell transcriptomic profiling of satellite glial cells directly involved in cardiac autonomic control – specifically in the stellate ganglion – was performed by van Weperen et al. (2021). SGCs were identified by high expression of glial-specific transcripts *S100b* and *Fabp7* and were shown to represent a fairly heterogeneous population. Five subpopulations were identified, representing different states of maturation and/or states of functionality. An immature cluster 1 was enriched in pluripotency markers *Ptprz1* and *Itgb8*. Maturation was associated with increasing metabolic functions, such as cholesterol syntheses (clusters 2–3), activation of cellular stress pathways (clusters 3–4) and, finally, senescence pathways (cluster 5, also characterised as quiescent or aged). As maturation progressed, the signalling pathway profile of SGCs also shifted – showing increases in aldosterone, endothelial nitric oxide synthase, purinergic and gap junction signalling (clusters 2–4), oxidative stress and, as a compensatory mechanism, melatonin signalling in cluster 5 (van Weperen et al., 2021).

These findings provide strong evidence that glial cells in the PNS are not homogeneous but exhibit functional specialisations that contribute directly to cardiac autonomic regulation.

**Role of glia within the autonomic cardiac ganglia in modulating heart rate and electrophysiology.** Not only are glial cells abundant in the CNS centres involved in autonomic cardiovascular regulation, where they play a crucial role in modulating cardiovascular reflexes (Mastitskaya et al., 2020), but peripheral glial cells also exert significant effects on autonomic nerve activity, in particular by directly regulating the electrical properties of neurons and cholinergic transmission in sympathetic ganglia (Enes et al., 2020). Selective activation of a Gq-coupled G protein-coupled receptor in GFAP$^+$ glial cells (astrocytes in CNS and non-myelinating glia in the PNS) *in vivo* increased heart rate and cardiac output

acutely and caused hypotension in the long-term through activation of sympathetic neurons in peripheral ganglia. This activation is most likely mediated by bidirectional purinergic mechanisms involving P2YRs (Xie et al., 2017).

Glial cells are considered a part of the cardiac ANS, which can be divided into extrinsic (associated with preganglionic axons) and intrinsic (ganglia and post-ganglionic axons) components (Fedele & Brand, 2020). Autonomic cardiac ganglia (ACG) regulate heart rate and cardiac output by serving as relay points for the ANS. ACG are primarily located in the epicardial fat within the transverse and oblique sinuses and around the roots of the great vessels such as the aorta and pulmonary arteries. The postganglionic axons from these ganglia extend to the SAN and AVN areas. PSNS and SNS in cardiac tissue also contain interconnecting neurons that integrate inputs for coordinated cardiac regulation (Coote, 2013).

Autonomic cardiac ganglia are made up of the following types of tissue:

- Nerve tissue. This includes sympathetic ganglia and nerve fibres that release noradrenaline, as well as parasympathetic fibres from the vagus nerve releasing acetylcholine (ACh). Interconnecting neurons facilitate communication between these inputs.
- Glial cells. SCs and SGCs support the neurons within the autonomic ganglia.
- Connective tissue. The fibrous skeleton and epicardial fat provide structural support and house the autonomic cardiac ganglia, offering protection and insulation.
- Cardiac myocytes (including pacemaker cells) and vascular tissue (generating nitric oxide and other neuro- and vasoactive substances contributing to metabolic support and regulation of the ganglia).

Kanemaru et al. (2023) reported on the *NC2_glial_NGF*$^+$ glial cell population serving as a niche partner for cardiac conduction system cells within the SAN, AVN and the bundle of His. These glial cells express key components necessary for maintaining the glutamine pool, potentially facilitating cardiac glutamatergic signalling in a manner analogous to astrocytes. Additionally, this analysis highlighted numerous trans-synaptic adhesion interactions, suggesting a synapse-like interconnection, as evidenced by the envelopment of pacemaker cells by glial processes. *NC2_glial_NGF*$^+$ cells may enhance cardiac conduction system innervation through the secretion of NGF (Kanemaru et al., 2023).

Moreover, the levels of production of BDNF and NGF, which are responsible for neuronal survival and differentiation, decrease with age (Elia et al., 2021). These factors in the PNS are primarily produced by glial cells such as SCs. Further, the numbers of glial cells in the PNS also decline. A retrospective analysis of 40 histological samples of ganglia from the epicardial ganglionated plexus

of the hearts from infants (9 days to 4 months), adult (31–44 years) and older people (61–78 years), concluded that the size of ganglia significantly increases with age, whereas the density of satellite cells, neuronal packing density and the area occupied by neuronal cell bodies in cardiac ganglia decrease in adult and aged humans compared to infants (Jurgaitiene et al., 2004).

The ACG provide communication between the ANS and the cardiovascular system. Sympathetic stimulation enhances SAN activity, increasing heart rate through the secretion of noradrenaline, which binds to $\beta$-adrenergic receptors on cardiac cells. This interaction activates adenylate cyclase, elevating cAMP levels and subsequently enhancing calcium influx through L-type channels. The resulting increase in calcium not only accelerates pacemaker potentials within the SAN but also amplifies myocardial contractility (Donald & Lakatta, 2023; Fozzard, 2002).

In contrast, the parasympathetic fibres primarily release ACh, which interacts with muscarinic M2-receptors on cardiac cells. This binding inhibits adenylate cyclase, curtails cAMP levels and thus diminishes calcium influx, leading to a deceleration of heart rate and reduction in the force of cardiac contraction. Additionally, ACh opens potassium channels, inducing hyperpolarization in pacemaker cells and further moderating the heart rate (Fozzard, 2002; Harvey & Belevych, 2003).

The network of interconnecting communications within the ACG modulates the balance between the SNS and PSNS by adapting neurotransmitter release in response to signals such as changes in blood pressure, which are detected by baroreceptors in the atrium (Guyenet, 2006).

SCs are vital in ensuring efficient propagation of electrical signals for maintaining rhythm and contractility (Jessen & Mirsky, 2005). Firstly, SCs in ACG provide myelination. This myelination ensures that the autonomic signals are transmitted swiftly and accurately, which is important for timely regulation of heart rate and contractility (Fields & Burnstock, 2006; Jessen & Mirsky, 2005). Secondly, glial cells in ACG modulate optimal neurotransmitter levels. For instance, they can uptake excess neurotransmitters such as noradrenaline and ACh from the synaptic cleft, thus preventing overstimulation (Fields & Burnstock, 2006; Volterra & Meldolesi, 2005). Further, glial cells regulate extracellular concentration of ions. For example, potassium channels on glial cells can uptake excess $K^+$ ions released during neuronal activity, preventing hyperkalaemia, which could disrupt the cardiac AP propagation and lead to arrhythmias. Additionally, glial cells express calcium-binding proteins and calcium pumps that help regulate $Ca^{2+}$ levels in the extracellular space (Hamilton & Attwell, 2010; Kléber & Rudy, 2004).

These data demonstrate how glial cells, specifically within the ACG, not only provide structural support, but also contribute to the modulation of neural activity through several mechanisms, such as the glia in the CNS.

Arrhythmias, such as atrial fibrillation (AF) and ventricular tachycardia, can occur when the balance of autonomic regulation and ion homeostasis is disrupted. Glial cells in the autonomic cardiac ganglia support neuronal activity that regulates heart rate and rhythm. They modulate neurotransmitter levels (Hanani & Spray, 2020), buffer extracellular ions and provide myelination for efficient signal transmission (Jessen & Mirsky, 2005). These functions are essential for stable autonomic input to the heart. Glial dysfunction, via impaired ion buffering or neuro-immune signalling, can contribute to maladaptive remodelling and arrhythmogenesis (Scherschel et al., 2019). While glial cells do not directly generate cardiac action potentials, the roles of glial cells in ion regulation, neurotransmitter uptake and myelination are indispensable for the electrophysiological stability of the heart (Kettenmann & Ransom, 2012).

A compelling demonstration of the functional significance of glia in cardiac electrophysiology comes from the work of Scherschel et al. (2019), who showed that activation of glial cells in the atrial ganglionated plexi contributes directly to AF. Their study revealed that catheter ablation, a common AF treatment, activates local glial cells, leading to the release of the calcium-binding protein S100B. This protein promotes sympathetic nerve sprouting and electrophysiological remodelling – factors increasing AF susceptibility (Cao et al., 2000). These findings underscore that glial cells are not passive structural elements but active participants in arrhythmogenesis through paracrine signalling. It further highlights the broader role of glial-derived molecules like S100B in modulating cardiac electrophysiology, especially under stress or disease conditions.

In addition to intrinsic cardiac ganglia, extrinsic autonomic ganglia such as the DRG, nodose ganglia, and stellate ganglia also contain glial cells that contribute to cardiac autonomic regulation, including cardiovascular reflexes.

In the DRG, SGCs encase sensory neurons transmitting pain and mechanosensory input. These glial cells express S100B, GFAP and purinergic receptors (P2X7, P2Y12) and regulate neurotransmitter levels, ion buffering and cytokine signalling under inflammatory conditions, indirectly influencing autonomic outflow (Hanani & Spray, 2020; Jessen & Mirsky, 2005).

The nodose ganglia, which house vagal afferent neurons projecting to the brainstem, are also surrounded by SGCs of neural crest origin. These cells help maintain synaptic stability and neurotransmitter balance, supporting vagal tone and reflex control of the heart rate (Hanani & Spray, 2020).

The stellate ganglia contain sympathetic neurons projecting to the heart and are supported by SGCs

expressing Kir4.1 and GFAP. These glial cells maintain the extracellular environment, modulate sympathetic excitability, and, when dysfunctional, are implicated in arrhythmogenesis (Li et al., 2025).

**Glial regulation of neurotransmission in the heart.** The role of glia in neurotransmission within the heart remains largely unexplored, however, investigating this area could yield important insights into cardiac neural regulation. Similar to the glial roles in the CNS, PNS glia play an active and essential role in modulating neurotransmission at the level of the neuromuscular junction (NMJ) (Ko & Robitaille, 2015). The NMJ consists of the presynaptic nerve terminal, postsynaptic muscle fibres and glial element – perisynaptic non-myelinating Schwann cells (PSCs) (Ko & Robitaille, 2015). PSCs play key roles in synapse formation, plasticity, maintenance and regeneration, forming the basis of the 'tripartite' synapse concept. Each NMJ has three to five PSCs, with their number correlating to endplate size. The PSCs express muscarinic (M1, M2, M5) and purinergic (adenosine A1, A2a) receptors and dynamically respond to synaptic transmission and modulate synaptic properties in physiological and pathological contexts via G-protein coupled receptors and $Ca^{2+}$ dependent mechanisms (Ko & Robitaille, 2015). This sensitivity can be reduced by substance P or NO during intense sustained stimulation, i.e. their response varies with activity patterns. For example, PSCs in soleus NMJ detect different patterns of synaptic activity and respond with a sustained oscillating $Ca^{2+}$ elevation and depression mediated via A1 receptors during bursting stimulations, while continuous stimulation generates large phasic $Ca^{2+}$ elevations and potentiation mediated via A2a receptors (Todd et al., 2010). In cardiac ganglia, adenosine acting on PSCs may contribute to its protective effects in cardiac ischaemia and reperfusion via presynaptic inhibition of noradrenaline release (Richardt et al., 1996). This presynaptic action of adenosine is, however, rapidly lost with ongoing ischaemia, while postsynaptic density and functional coupling of adenosine receptors is preserved during ischaemia, and endogenous adenosine modulates catecholamine-mediated responses of ischaemic myocardium such as ventricular arrhythmias (Lerman, 1993).

It is important to note that the concept of the tripartite synapse (bidirectional communication between astrocytes and neurons) is well studied for the CNS, while the role of glia in neural transmission in the PNS has only recently started attracting attention (Tedoldi et al., 2021).

Satellite glial cells in autonomic ganglia are known to buffer and uptake excess neurotransmitters, in particular glutamate, thus preventing neuronal damage and death via excitotoxicity (Martineau et al., 2020). Also, satellite glia buffer extracellular potassium and calcium levels, thus maintaining the ionic environment (Hanani &

Spray, 2020). When the buffering capacity of satellite glia in autonomic ganglia is compromised, for example in inflammatory conditions, this may contribute to peripheral neurodegeneration and impaired synaptic transmission in cardiac autonomic ganglia, just like in the CNS (Yang & Zhou, 2019).

In addition to their buffering functions, glial cells also participate in gliotransmission – the regulated storage and release of neurotransmitters – enabling direct modulation of synaptic activity and glial–neuronal communication. They further support neuronal development and synaptic plasticity by producing neurotrophic factors:

NGF. Promotes neuronal differentiation (Zhao et al., 2024).

BDNF. Acts on neurons, leading to increased intracellular $Ca^{2+}$ levels and activation of cAMP response element-binding protein, which encourages the growth of new synapses (Bathina & Das, 2015). Low circulating BDNF correlates with elevated NT-proBNP and adverse cardiac remodelling, indicating its cardioprotective potential (Bahls et al., 2019).

GDNF. Responsible for neuronal survival, especially of the dopaminergic type (Chinta & Andersen, 2005; Granholm et al., 2000) and plays a fundamentally important role in the PNS where it promotes muscle trophism, presynaptic maturation and acts as a neuromodulator of synaptic transmission (Stanga et al., 2021).

In summary, glial cells are indispensable for the regulation of neurotransmitter release and reuptake through a variety of molecular mechanisms. They maintain synaptic homeostasis by modulating neurotransmitter levels, providing metabolic support and releasing neurotrophic factors that influence synaptic plasticity and neuronal survival. This multifaceted role underscores the critical importance of glial cells in both CNS and PNS function, highlighting their potential as therapeutic targets in neurodegenerative diseases and peripheral neuropathies often affecting the heart in cardiometabolic disease (e.g. cardiac autonomic neuropathy in diabetes).

## Role of cardiac glia in disease

In the heart, glial cells such as SCs and CNG contribute to the regulation of heart rate, synaptic transmission, structural support and the development of cardiac innervation (Elia & Fossati, 2023; Kikel-Coury et al., 2021).

SCs in the PNS provide myelination to axons, ensuring efficient signal transmission, which is essential for proper cardiac function. They also respond to injury by promoting repair and modulating the inflammatory response. Dysfunction in these cells can lead to impaired

signal transmission and contribute to arrhythmias (Guyenet, 2006; Jessen & Mirsky, 2005).

CNGs, which resemble astrocytes in the CNS, are involved in maintaining the homeostasis of the cardiac microenvironment. They regulate neurotransmitter levels, clear cellular debris and modulate synaptic plasticity (Kikel-Coury et al., 2021).

Glial cells influence various cardiac conditions through several mechanisms:

Myocardial infarction. During myocardial infarction the clearance of dead cells and debris by glial cells can significantly affect the inflammatory response and subsequent tissue repair processes. Efficient phagocytosis by glial cells helps to mitigate excessive inflammation and promotes healing (Metcalf et al., 2024).

Cardiomyopathy and heart failure. Glial dysfunction can contribute to maladaptive cardiac remodelling. The inability to adequately clear cellular debris and regulate inflammation can exacerbate cardiac dysfunction and progression of the disease (Kikel-Coury et al., 2021). In patients with cardiomyopathy and refractory ventricular arrhythmia, glial cell activation in stellate ganglia contributes to inflammation and neurochemical remodelling and is thus likely contribute to excessive and dysfunctional sympathetic tone (Ajijola et al., 2017).

Arrhythmias. SCs ensure proper electrical conduction in the heart. Dysfunction in these cells can lead to impaired myelination and signal transmission, increasing the risk of arrhythmias. By modulating ion homeostasis and neurotransmitter levels, glial cells help maintain the electrophysiological stability of the heart (Zhang et al., 2010).

Targeting glial cells offers a promising therapeutic approach for various cardiac diseases. Enhancing the phagocytic activity of glial cells could improve the clearance of apoptotic cells and reduce inflammation, promoting tissue repair and preventing further damage (Metcalf et al., 2024). Additionally, modulating glial cell function to prevent excessive inflammation and fibrosis could be beneficial in treating chronic cardiac conditions.

An analysis of transcriptome for astrocytes and neurons demonstrated that CNS glial cells in *Drosophila* brain and *Caenorhabditis elegans* are similar to mammalian astrocytes and functionally responsible for cleaning apoptotic cells by phagocytosis. This supports the idea that glial phagocytic pathways participate in CNS and PNS synapse elimination, amyloid clearance within the brain and axonal pruning (Cahoy et al., 2008).

Metcalf et al. (2024) revealed that glial cells exert significant influence over organismal autophagy and lipid metabolism through non-cell-autonomous mechanisms. The research highlights how the expression of the transcription factor X-box binding protein 1 (which regulates genes involved in protein folding, secretion,

degradation and lipid biosynthesis) in glial cells enhances proteostasis and longevity by reprogramming lipid metabolism and activating autophagy in other cells, including neurons. This interaction underscores the key role of glial cells in maintaining metabolic homeostasis and offers potential therapeutic targets for metabolic and neurodegenerative diseases (Metcalf et al., 2024).

By understanding the mechanisms of glial cell activity, particularly their roles in autophagy, lipid metabolism and debris clearance, we can gain insight into the pathology of diseases associated not only with CNS neurodegeneration such as Alzheimer's and Parkinson's, but also peripheral neurodegeneration and neuropathies, potentially offering new therapeutic targets for treatment of cardiac autonomic neuropathy in metabolic disease.

## Conclusion

Glial cells, once considered merely supportive elements, are now recognized as active regulators of both neural and cardiac health. In the heart, glial cells contribute not only to structural support but also to the fine-tuning of heart rate and synaptic transmission. Schwann cells preserve innervation and fascilitate repair, satellite glial cells modulate inflammatory responses, provide metabolic support and regulate neurotransmitter and ion homeostasis, while cardiac nexus glia modulate sympathetic and parasympathetic input to maintain autonomic balanc. The involvement of glial cells in key processes such as ion homeostasis, phagocytosis and inflammation is increasingly appreciated for its impact on cardiac pathophysiology. Efficient phagocytosis by glia mitigates inflammation and supports tissue repair following myocardial infarction, whereas glial dysfunction may contribute to the progression of cardiomyopathy and heart failure by disrupting electrophysiological stability. Despite these advances, important questions remain regarding the precise molecular mechanisms underlying glial modulation of cardiac function and how glial dysfunction drives disease progression. Future research should focus on elucidating glial cell signalling pathways, their interactions with cardiac neurons, and their role in cardiac remodelling. Such insights could pave the way for novel therapeutic interventions targeting glial cells to improve outcomes in cardiac diseases.

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

## Additional information

### Competing interests

The authors declare they have no competing interests.

### Author contributions

All authors have approved the final version of the manuscript and agree to be accountable for all aspects of the work in ensuring that questions related to the accuracy or integrity of any part of the work are appropriately investigated and resolved. All persons designated as authors qualify for authorship, and all those who qualify for authorship are listed.

### Funding

This work was funded by the British Heart Foundation (BHF), FS/IBSRF/21/25060 to S.M.

### Keywords

cardiac glial cells, cardiac nexus glia, satellite glial cells, Schwann cells

### Supporting information

Additional supporting information can be found online in the Supporting Information section at the end of the HTML view of the article. Supporting information files available:

**Peer Review History**

