## [Peer Review History · The Journal of Physiology]

Glial cells in the heart: implications for their roles in health and disease

Svetlana Mastitskaya, Rimma Dugarova, and Shefeeq M Theparambil

DOI: 10.1113/JP286598

Corresponding author(s): Svetlana Mastitskaya (svetlana.mastitskaya@bristol.ac.uk)

Review Timeline:

Submission Date:	25-Feb-2025
Editorial Decision:	22-Apr-2025
Revision Received:	30-Jun-2025
Editorial Decision:	21-Jul-2025
Revision Received:	08-Aug-2025
Accepted:	15-Aug-2025

Senior Editor: Bjorn Knollmann

Reviewing Editor: T Alexander Quinn

Transaction Report:

Dear Dr Mastitskaya,

Re: JP-TR-2025-286598 "Glial cells in the heart and their roles in health and disease" by Rimma Dugarova, Shefeeq M Theparambil, and Svetlana Mastitskaya

Thank you for submitting your manuscript to The Journal of Physiology. It has been assessed by a Reviewing Editor and by 2 expert referees and we are pleased to tell you that it is potentially acceptable for publication following satisfactory major revision.

Please address all the points raised and incorporate all requested revisions or explain in your Response to Referees why a change has not been made. We hope you will find the comments helpful and that you will be able to return your revised manuscript within 2 months. If you require longer than this, please contact journal staff: jp@physoc.org. Please note that this letter does not constitute a guarantee for acceptance of your revised manuscript.

ABSTRACT FIGURES: Authors are expected to use The Journal's premium BioRender account to create/redraw their Abstract Figures. Information on how to access this account is here:

<https://physoc.onlinelibrary.wiley.com/journal/14697793/biorender-access>.

REVISION CHECKLIST:

IMPORTANT POINTS TO NOTE WHEN REVISING YOUR MANUSCRIPT:

LANGUAGE EDITING AND SUPPORT FOR PUBLICATION: If you would like help with English language editing, or other article preparation support, Wiley Editing Services offers expert help, including English Language Editing, as well as translation, manuscript formatting, and figure formatting at www.wileyauthors.com/eeo/preparation. You can also find resources for Preparing Your Article for general guidance about writing and preparing your manuscript at www.wileyauthors.com/eeo/prepareresources.

We look forward to receiving your revised submission.

Yours sincerely,

Bjorn Knollmann
Senior Editor
The Journal of Physiology

REQUIRED ITEMS

- Please include an Abstract Figure file, as well as the Figure Legend text within the main article file. The Abstract Figure is a piece of artwork designed to give readers an immediate understanding of the Review Article and should summarise the main conclusions. If possible, the image should be easily 'readable' from left to right or top to bottom. It should show the physiological relevance of the Review so readers can assess the importance and content of the article. Abstract Figures should not merely recapitulate other figures in the Review. Please try to keep the diagram as simple as possible and without superfluous information that may distract from the main conclusion of the Review. Abstract Figures must be provided by authors no later than the revised manuscript stage and should be uploaded as a separate file during online submission labelled as File Type 'Abstract Figure'. Please ensure that you include the figure legend in the main article file. All Abstract Figures will be sent to a professional illustrator for redrawing and you may be asked to approve the redrawn figure before your paper is accepted.

- Your MS must include a complete "Additional information section" with the following 4 headings and content:

Competing Interests: A statement regarding competing interests. If there are no competing interests, a statement to this effect must be included. All authors should disclose any conflict of interest in accordance with journal policy.

Author contributions: Each author should take responsibility for a particular section of the study and have contributed to writing the paper. Acquisition of funding, administrative support or the collection of data alone does not justify authorship; these contributions to the study should be listed in the Acknowledgements. Additional information such as 'X and Y have contributed equally to this work' may be added as a footnote on the title page.

It must be stated that all authors approved the final version of the manuscript and that all persons designated as authors qualify for authorship, and all those who qualify for authorship are listed.

Funding: Authors must indicate all sources of funding, including grant numbers. If authors have not received funding, this must be stated.

It is the responsibility of authors funded by RCUK to adhere to their policy regarding funding sources and underlying research material. The policy requires funding information to be included within the acknowledgement section of a paper. Guidance on how to acknowledge funding information is provided by the Research Information Network. The policy also requires all research papers, if applicable, to include a statement on how any underlying research materials, such as data, samples or models, can be accessed. However, the policy does not require that the data must be made open. If there are considered to be good or compelling reasons to protect access to the data, for example commercial confidentiality or legitimate sensitivities around data derived from potentially identifiable human participants, these should be included in the statement.

Acknowledgements: Acknowledgements should be the minimum consistent with courtesy. The wording of acknowledgements of scientific assistance or advice must have been seen and approved by the persons concerned. This section should not include details of funding.

- Please upload separate high quality figure files via the submission form.

- Author profile(s) must be uploaded via the submission form. Authors should submit a short biography (no more than 100 words for one author or 150 words in total for two authors) and a portrait photograph of the two leading authors on the paper. These should be uploaded and clearly labelled together in a Word document with the revised version of the manuscript. Any standard image format for the photograph is acceptable, but the resolution should be at least 300 DPI and

preferably more. A group photograph of all authors is also acceptable, providing the biography for the whole group does not exceed 150 words.

- Please include a full title page as part of your main article (Word) file, which should contain the following: title, authors, affiliations, corresponding author name and contact details, keywords, and running title.

EDITOR COMMENTS

Reviewing Editor:

Your paper has been reviewed by two experts in the field, who both felt it was interesting, timely, and highlights a burgeoning area of research. However, as described in their comments below, the paper needs revision to improve its focus and potential impact. In particular, the text should be revised so that statements and conclusions are better supported by reported evidence and the focus is more specific to the heart. In addition, the figures need more depth and additional summary tables should be included. Please revise the manuscript accordingly, including a point-by-point response to the reviewers' suggestions.

Please also see 'Required Items' above.

Senior Editor:

I concur with the reviewing editor's recommendation

REFEREE COMMENTS

Referee #1:

This review is overall generally interesting, and frames glial cell development and function in health and disease from a cardiac standpoint. While glial cells in the heart are poorly understood and there is much to be learned about them, this review does a fair job summarizing what is known.

Suggestions for improving the review are below:

- The review references cardiac autonomic ganglia but appears to mean epicardial ganglia or ganglionated plexi. If the authors use the latter terminology, their description of epicardial ganglia would be appropriate. If the authors wish to discuss cardiac autonomic ganglia, then glia within the dorsal root ganglia, nodose ganglia, and stellate ganglia must be included.
- Reference for Panel B in summary figure should be provided.
- There is no real mention of gliotransmission in the review, this is an important aspect of glial-glia and glial neuronal communication that would be important to discuss.
- The summary figure could use more depth to make it a citation classic or an image used by the readership ... it would encourage citation of this review as well.
- A table summarizing similarities/differences between the various glial populations across "autonomic ganglia" would be useful.
- There are publications on the transcriptome of glial cells in the PNS, which sheds more light on their functional roles in cardiac neural control. Such data should be included here.

Referee #2:

The topical review article "Glial cells in the heart and their roles in health and disease" by Dugarova et al. provides a timely highlight for this research area. The review provides a covering of the research to date and acknowledgment of its paucity. There is good description of the relevance of glial cells in the central nervous system, from development through to disease, and comparative description of knowledge of these same areas in the heart. The figures and table provide good support for the text.

There are places in the review that read more of an opinion piece than a review of the literature. The reviewer acknowledges that the low amount of published work in this field is limiting for an in-depth review, however a more concise review would improve the manuscript. This is especially relevant in Section 2.1 regarding reinnervation of the heart where the roles of glia are largely postulated rather than summarized research findings. Other areas that are published deserve further in-depth attention, for example the S100B and AF findings from Scherschel et al.

Section 1.2. Please provide references for the statements "...by a gap of about 20 nm" and "Interactions between SGCs and neurons occur via ionic channels and mediators" and provide further description of this work and the methods and data showing this. In addition, in Section 3.1 (lines 510-511) it is stated about glial cells in the ACG that "They maintain the proper timing of cardiac contractions and prevent the erratic electrical arrhythmic signaling". This is a major conclusion and it is not clear what data this is based on. Please provide references and a description of the published data.

Section 3.2 contains significant review of the roles of glia at the neuromuscular junction. This could be reduced so the focus remains on cardiac glia. It is not clear what systems are summarized in lines 555-562, these need to be integrated into the text to clarify relevance.

The conclusion section reads as a list. It would be helpful to integrate these into a cohesive conclusion section and highlight here the areas of further study needed.

There are many abbreviations used in the review that are not well known. I suggest a list of abbreviations be provided.

The article is let down by numerous grammatical, spelling and language errors. Examples are: lines 55, 76-77, 129, 130, 135, 169, 210, 250-251, 262, 271, 272, 430. There are likely others and the authors needs to undertake a complete review of the text and correct these.

END OF COMMENTS

The topical review article “Glial cells in the heart and their roles in health and disease” by Dugarova et al. provides a timely highlight for this research area. The review provides a covering of the research to date and acknowledgment of its paucity. There is good description of the relevance of glial cells in the central nervous system, from development through to disease, and comparative description of knowledge of these same areas in the heart. The figures and table provide good support for the text.

There are places in the review that read more of an opinion piece than a review of the literature. The reviewer acknowledges that the low amount of published work in this field is limiting for an in-depth review, however a more concise review would improve the manuscript. This is especially relevant in Section 2.1 regarding reinnervation of the heart where the roles of glia are largely postulated rather than summarised research findings. Other areas that are published deserve further in depth attention, for example the S100B and AF findings from Scherschel et al.

Section 1.2. Please provide references for the statements “...by a gap of about 20 nm” and “Interactions between SGCs and neurons occur via ionic channels and mediators” and provide further description of this work and the methods and data showing this. In addition, in Section 3.1 (lines 510-511) it is stated about glial cells in the ACG that “They maintain the proper timing of cardiac contractions and prevent the erratic electrical arrhythmic signaling”. This is a major conclusion and it is not clear what data this is based on. Please provide references and a description of the published data.

Section 3.2 contains significant review of the roles of glia at the neuromuscular junction. This could be reduced so the focus remains on cardiac glia. It is not clear what systems are summarised in lines 555-562, these need to be integrated into the text to clarify relevance.

The conclusion section reads as a list. It would be helpful to integrate these into a cohesive conclusion section and highlight here the areas of further study needed.

There are many abbreviations used in the review that are not well known. I suggest a list of abbreviations be provided.

The article is let down by numerous grammatical, spelling and language errors. Examples are: lines 55, 76-77, 129, 130, 135, 169, 210, 250-251, 262, 271, 272, 430. There are likely others and the authors needs to undertake a complete review of the text and correct these.

MS # JP-TR-2025-286598
Responses to referees' comments

We are grateful for the constructive comments of both Reviewers and Reviewing Editor of our original submission to the *J Physiology* and have taken full account of the raised criticisms. We are delighted to have an opportunity to re-submit our work. We now revised the text so that the statements and conclusions are more supported by the evidence and focused more specifically on the heart. We also include improved quality figures requested by the reviewers and provide a full response to their comments as well as a thoroughly revised manuscript.

Below we state the criticisms ("critique") and then provide our responses.

Referee #1

This review is overall generally interesting, and frames glial cell development and function in health and disease from a cardiac standpoint. While glial cells in the heart are poorly understood and there is much to be learned about them, this review does a fair job summarizing what is known. Suggestions for improving the review are below.

Response: We would like to thank this referee for taking time to review our manuscript and for their overall positive assessment of our work. We have now included additional figures, provided our responses to all the criticisms raised and submitted a thoroughly revised manuscript.

Critique:

The review references cardiac autonomic ganglia but appears to mean epicardial ganglia or ganglionated plexi. If the authors use the latter terminology, their description of epicardial ganglia would be appropriate. If the authors wish to discuss cardiac autonomic ganglia, then glia within the dorsal root ganglia, nodose ganglia, and stellate ganglia must be included.

Response: We thank the reviewer for this valuable observation. In response, we have revised the terminology throughout the manuscript to consistently use "epicardial ganglia" or "ganglionated plexi" when referring to the intrinsic cardiac autonomic network. This more accurately reflects the anatomical focus of our review. Additionally, to acknowledge the broader scope of cardiac autonomic regulation, we have expanded Section 3.2 to include a new paragraph discussing glial populations within key extrinsic autonomic ganglia – namely, the dorsal root ganglia, nodose ganglia and stellate ganglia. This addition helps situate epicardial glial populations within the larger context of peripheral autonomic control.

Critique:

Reference for Panel B in summary figure should be provided.

Response: Panel B of the summary figure was created by the authors using BioRender.com. We have now clarified this in the figure legend by adding the statement: "Created with BioRender".

Critique:

There is no real mention of gliotransmission in the review, this is an important aspect of glial-glia and glial neuronal communication that would be important to discuss.

Response: Thank you for this suggestion. Gliotransmission is indeed relatively well studied in brain glial cells and, to some extent, at neuromuscular junctions. However, this aspect has not been extensively addressed in cardiac glial cells. We have now incorporated this topic into Section 3.3, where we discuss gliotransmission and its role

in glial-glial and glial-neuronal communication within cardiac autonomic regulation. We have also amended the section title for better clarity.

Critique:

The summary figure could use more depth to make it a citation classic or an image used by the readership ... it would encourage citation of this review as well.

Response: Thank you for the valuable comment regarding the summary figure. We have substantially revised Figure 1 to add more depth and integrative elements. The updated figure now illustrates both the sympathetic (SNS) and parasympathetic (PSNS) components of the cardiac autonomic nervous system, including anatomical pathways with pre- and postganglionic neurons, neurotransmitter types and their effects on heart rate and contractility. We believe the revised figure enhances the visual clarity of the review.

Critique:

A table summarizing similarities/differences between the various glial populations across "autonomic ganglia" would be useful.

Response: Thank you for this suggestion. We have now incorporated a table explaining the different glial population and their characteristics in autonomic ganglia (Table 1).

Critique:

There are publications on the transcriptome of glial cells in the PNS, which sheds more light on their functional roles in cardiac neural control. Such data should be included here.

Response: Thank you for pointing out the importance of incorporating transcriptomic data on glial cells in the peripheral nervous system. In response, we have integrated this content into the revised manuscript (Section 3.1), highlighting the relevance of these findings to cardiac autonomic regulation and further supporting the view of glial cells as active participants in heart neurophysiology.

Referee #2:

The topical review article "Glial cells in the heart and their roles in health and disease" by Dugarova et al. provides a timely highlight for this research area. The review provides a covering of the research to date and acknowledgment of its paucity. There is good description of the relevance of glial cells in the central nervous system, from development through to disease, and comparative description of knowledge of these same areas in the heart. The figures and table provide good support for the text.

Response: We would like to thank the referee for taking the time to review our manuscript and for their overall positive assessment of our work. We now provide our responses to all the criticisms raised and submit a thoroughly revised manuscript.

Critique 1: There are places in the review that read more of an opinion piece than a review of the literature. The reviewer acknowledges that the low amount of published work in this field is limiting for an in-depth review, however a more concise review would improve the manuscript. This is especially relevant in Section 2.1 regarding reinnervation of the heart where the roles of glia are largely postulated rather than summarized research findings. Other areas that are published deserve further in-depth attention, for example the S100B and AF findings from Scherschel et al.

Response: We thank the reviewer for this important observation. In response, we have expanded our discussion in Section 3.2 to more thoroughly describe the findings of Scherschel et al. (2019) regarding S100B-mediated glial contributions to atrial

fibrillation, as suggested. This study is also mentioned in section 1.3. Additionally, we have amended the Section 2.1 to include the relevant citations for all statements and to make it more evidence-based.

Critique 2: Section 1.2. Please provide references for the statements "...by a gap of about 20 nm" and "Interactions between SGCs and neurons occur via ionic channels and mediators" and provide further description of this work and the methods and data showing this. In addition, in Section 3.1 (lines 510-511) it is stated about glial cells in the ACG that "They maintain the proper timing of cardiac contractions and prevent the erratic electrical arrhythmic signaling". This is a major conclusion and it is not clear what data this is based on. Please provide references and a description of the published data.

Response: We thank the reviewer for raising these points. We have completely rewritten the Section 1.2 and added relevant citations. The statement in the former Section 3.1 (now 3.2) regarding glial cells in the ACG maintaining the timing of contractions and preventing erratic electrical signaling has also been revised, with appropriate references now provided.

Critique 3: Section 3.2 contains significant review of the roles of glia at the neuromuscular junction. This could be reduced so the focus remains on cardiac glia. It is not clear what systems are summarized in lines 555-562, these need to be integrated into the text to clarify relevance.

Response: Thank you for these comments. We have shortened the section on NMJ; however, information on the role of glia in cardiac NMJ is very limited, so we have attempted to extrapolate from the existing evidence on skeletal NMJ. Additionally, we have revised the part summarising the neurotrophic factors produced by glia (formerly lines 555-562) for better clarity.

Critique 4: The conclusion section reads as a list. It would be helpful to integrate these into a cohesive conclusion section and highlight here the areas of further study needed.

Response: Thank you for this important point. We have now updated the conclusion in the revised manuscript.

Critique 5: There are many abbreviations used in the review that are not well known. I suggest a list of abbreviations be provided.

Response: We have now provided a list of abbreviations.

Critique 6: The article is let down by numerous grammatical, spelling and language errors. Examples are: lines 55, 76-77, 129, 130, 135, 169, 210, 250-251, 262, 271, 272, 430. There are likely others and the authors needs to undertake a complete review of the text and correct these.

Response: Thank you very much for pointing out these typos and grammatical inconsistencies. We have now thoroughly revised the text.

Dear Dr Mastitskaya,

Re: JP-TR-2025-286598R1 "Glial cells in the heart and their roles in health and disease" by Svetlana Mastitskaya, Rimma Dugarova, and Shefeeq M Theparambil

Thank you for submitting your revised manuscript to The Journal of Physiology. It has been assessed by a Reviewing Editor and by 2 expert referees and we are pleased to tell you that it is acceptable for publication following satisfactory revision.

ABSTRACT FIGURES: Authors may use The Journal's premium BioRender account to create/redraw their Abstract Figures (and any other suitable schematic figure). Information on how to access this account is here: <https://physoc.onlinelibrary.wiley.com/journal/14697793/biorender-access>.

REVISION CHECKLIST: Upload a full Response to Referees file. To create your 'Response to Referees' copy all the reports, including any comments from the Senior and Reviewing Editors, into a Microsoft Word, or similar, file and respond to each point, using font or background colour to distinguish comments and responses and upload as the required file type.

We look forward to receiving your revised submission.

Yours sincerely,

Bjorn Knollmann
Senior Editor

REQUIRED ITEMS

- Please include the Figure Abstract Figure Legend text within the main article file.

- Your MS must include a complete "Additional information section" with the following 4 headings and content:

Competing Interests: A statement regarding competing interests. If there are no competing interests, a statement to this effect must be included. All authors should disclose any conflict of interest in accordance with journal policy.

Author contributions: Each author should take responsibility for a particular section of the study and have contributed to writing the paper. Acquisition of funding, administrative support or the collection of data alone does not justify authorship; these contributions to the study should be listed in the Acknowledgements. Additional information such as 'X and Y have contributed equally to this work' may be added as a footnote on the title page.

It must be stated that all authors approved the final version of the manuscript and that all persons designated as authors qualify for authorship, and all those who qualify for authorship are listed.

Funding: Authors must indicate all sources of funding, including grant numbers. If authors have not received funding, this must be stated.

It is the responsibility of authors funded by RCUK to adhere to their policy regarding funding sources and underlying research material. The policy requires funding information to be included within the acknowledgement section of a paper. Guidance on how to acknowledge funding information is provided by the Research Information Network. The policy also requires all research papers, if applicable, to include a statement on how any underlying research materials, such as data, samples or models, can be accessed. However, the policy does not require that the data must be made open. If there are considered to be good or compelling reasons to protect access to the data, for example commercial confidentiality or legitimate sensitivities around data derived from potentially identifiable human participants, these should be included in the statement.

Acknowledgements: Acknowledgements should be the minimum consistent with courtesy. The wording of acknowledgements of scientific assistance or advice must have been seen and approved by the persons concerned. This section should not include details of funding.

EDITOR COMMENTS

Reviewing Editor:

The authors have addressed much of the reviewers' concerns and in doing so have strengthened the paper.

The second reviewer, however, still has valid concerns about the robustness of the review in some places, which makes this paper not yet acceptable in its current form.

This relates to some of the conclusions drawn in the paper and how the supporting evidence is presented, particularly when the referenced data is from a different organ system. The authors need to be very clear about what has and has not actually been directly shown in the heart. Related to this, the reviewer have also suggested a change of the title, to better reflect the current state of the field.

The reviewer also highlighted multiple instances where general reviews are cited for descriptions of specific findings and conclusions, which should instead describe and reference the original research papers. They also pointed out instances where the original paper has not been referenced or conclusions drawn from previous reports are not in line with the original findings.

With a further revision addressing these continuing concerns, this review should provide an important highlight of the current status of the field and where further work is needed.

Senior Editor:

While improved, I concur with the concerns raised by reviewer 2 and the reviewing editor. I encourage the authors to tone their general statements. Rather than overstating the role of glial cells based on non-cardiac studies, I suggest pointing the reader to open questions on the role of glial cells in the heart that deserve further experimentation. I also ask the authors to revise the title accordingly.

REFEREE COMMENTS

Referee #1:

The authors have adequately addressed my major comments.

Referee #2:

Manuscript "Glial cells in the heart and their roles in health and disease".

Thank you for the opportunity to re-review this manuscript, and for the authors revisions. Overall the revisions have strengthened the content, and the inclusion of Table 1 and associated references is especially helpful.

I do still hold some concern for the paucity of data versus the strength of conclusions made in this manuscript however. It would be helpful when something has not been specifically shown in the heart that this needs to be directly stated, rather than inferring roles from glial cells in the CNS or other regions of the PNS. I also suggest that the title also be updated to reflect the need for future work and data to be able to address their importance. A suggestion is "Glial cells in the heart: implications for their roles in health and disease".

There are multiple examples where general reviews are cited after descriptions of specific data findings and conclusions. For example, in Section 1.1, it is stated:

"SCs contribute to cardiac repair by supporting the survival and regeneration of cardiac neurons. They are involved in forming protective myelin sheaths around cardiac autonomic neurons, ensuring proper signal transmission that regulates heart rate and rhythm (Hadaya & Ardell, 2020). In response to cardiac injury, SCs can exhibit plasticity, similar to their response in peripheral nerve injury, aiding in the repair and regeneration of damaged cardiac tissue (Jessen & Mirsky, 2019)."

And

"For instance, SC-derived neuregulin-1 has been shown to promote cardiac repair and improve outcomes following myocardial infarction by enhancing cardiomyocyte survival and function (Galindo et al., 2014). This highlights the potential therapeutic applications of targeting SCs in cardiac repair strategies."

And

"Within cardiac ganglia, SGCs form functional networks around neurons and respond to microenvironmental changes, including inflammation or injury, by releasing signalling molecules such as ATP, glutamate, and cytokines. This bidirectional

signalling modulates autonomic control of cardiac function, facilitating physiological adaptation. Disrupted SGC-neuron communication during inflammation has been linked to cardiac dysregulation and arrhythmogenic pathologies (Jessen & Mirsky, 2005; Hanani & Spray, 2020)"

And

"Arrhythmias. SCs ensure proper electrical conduction in the heart. Dysfunction in these cells can lead to impaired myelination and signal transmission, increasing the risk of arrhythmias. By modulating ion homeostasis and neurotransmitter levels, glial cells help maintain the electrophysiological stability of the heart (Jessen & Mirsky, 2005; Guyenet, 2006; Kettenmann & Ransom, 2012)"

The references provided are general reviews. Providing the description from the original papers detailing these findings would strengthen the review.

The first evidence that identified glial cells in the heart and coined the name term cardiac nexus glia (CNG), also known as cardiac glial cells or cardiac Schwann cells, was published in 2021 (Kikel-Coury et al., 2021).

Please correct/clarify this statement as glial cells were identified in the heart by electron microscopy more than 40 years ago. The original papers should be described and cited.

The conclusion of the heart transplant and reinnervation section (raised as a concern in the first round of review as roles of glia were largely postulated rather than summarized research findings) requires further strengthening. Currently it appears factually stating "This multifaceted support from glial cells ensures that regenerating neurons can form functional synapses and integrate into existing neural circuits, facilitating the reinnervation of the heart after transplantation (Feng et al., 2012)." It is not clear how the paper cited is relevant to this statement as the paper is an in vitro paper with no research in heart reinnervation. It would be helpful to clarify the need to provide evidence for synaptic integration and reinnervation facilitation.

END OF COMMENTS

Manuscript “Glial cells in the heart and their roles in health and disease”.

Thank you for the opportunity to re-review this manuscript, and for the authors revisions. Overall the revisions have strengthened the content, and the inclusion of Table 1 and associated references is especially helpful.

I do still hold some concern for the paucity of data versus the strength of conclusions made in this manuscript however. It would be helpful when something has not been specifically shown in the heart that this needs to be directly stated, rather than inferring roles from glial cells in the CNS or other regions of the PNS. I also suggest that the title also be updated to reflect the need for future work and data to be able to address their importance. A suggestion is “Glial cells in the heart: implications for their roles in health and disease”.

There are multiple examples where general reviews are cited after descriptions of specific data findings and conclusions. For example, in Section 1.1, it is stated:

“SCs contribute to cardiac repair by supporting the survival and regeneration of cardiac neurons. They are involved in forming protective myelin sheaths around cardiac autonomic neurons, ensuring proper signal transmission that regulates heart rate and rhythm (Hadaya & Ardell, 2020). In response to cardiac injury, SCs can exhibit plasticity, similar to their response in peripheral nerve injury, aiding in the repair and regeneration of damaged cardiac tissue (Jessen & Mirsky, 2019).”

And

“For instance, SC-derived neuregulin-1 has been shown to promote cardiac repair and improve outcomes following myocardial infarction by enhancing cardiomyocyte survival and function (Galindo et al., 2014). This highlights the potential therapeutic applications of targeting SCs in cardiac repair strategies.”

And

“Within cardiac ganglia, SGCs form functional networks around neurons and respond to microenvironmental changes, including inflammation or injury, by releasing signalling molecules such as ATP, glutamate, and cytokines. This bidirectional signalling modulates autonomic control of cardiac function, facilitating physiological adaptation. Disrupted SGC–neuron communication during inflammation has been linked to cardiac dysregulation and arrhythmogenic pathologies (Jessen & Mirsky, 2005; Hanani & Spray, 2020)”

And

“Arrhythmias. SCs ensure proper electrical conduction in the heart. Dysfunction in these cells can lead to impaired myelination and signal transmission, increasing the risk of arrhythmias. By modulating ion homeostasis and neurotransmitter levels, glial cells help maintain the electrophysiological stability of the heart (Jessen & Mirsky, 2005; Guyenet, 2006; Kettenmann & Ransom, 2012)”

The references provided are general reviews. Providing the description from the original papers detailing these findings would strengthen the review.

The first evidence that identified glial cells in the heart and coined the name term cardiac nexus glia (CNG), also known as cardiac glial cells or cardiac Schwann cells, was published in 2021 (Kikel-Coury et al., 2021).

Please correct/clarify this statement as glial cells were identified in the heart by electron microscopy more than 40 years ago. The original papers should be described and cited.

The conclusion of the heart transplant and reinnervation section (raised as a concern in the first round of review as roles of glia were largely postulated rather than summarized research

findings) requires further strengthening. Currently it appears factually stating “*This multifaceted support from glial cells ensures that regenerating neurons can form functional synapses and integrate into existing neural circuits, facilitating the reinnervation of the heart after transplantation (Feng et al., 2012).*” It is not clear how the paper cited is relevant to this statement as the paper is an in vitro paper with no research in heart reinnervation. It would be helpful to clarify the need to provide evidence for synaptic integration and reinnervation facilitation.

MS # JP-TR-2025-286598R1
Responses to referee's comments

We are grateful for the constructive comments of the Reviewer 2 and the Reviewing Editor of our revised submission to the *J Physiology* and have taken full account of the raised criticisms. We have now revised the text to moderate our conclusions and to clearly indicate where evidence in the heart is lacking and where data are inferred from other organs. Below we state the criticisms ("critique") and then provide our responses.

Referee #2

Thank you for the opportunity to re-review this manuscript, and for the authors' revisions. Overall the revisions have strengthened the content, and the inclusion of Table 1 and associated references is especially helpful. I do still hold some concern for the paucity of data versus the strength of conclusions made in this manuscript, however. It would be helpful when something has not been specifically shown in the heart that this needs to be directly stated, rather than inferring roles from glial cells in the CNS or other regions of the PNS. I also suggest that the title also be updated to reflect the need for future work and data to be able to address their importance. A suggestion is "Glial cells in the heart: implications for their roles in health and disease".

Response: We would like to thank the referee for taking the time to review our manuscript again, and for their constructive criticism and overall positive assessment of our work. We are especially grateful for the suggestions regarding a new title and the importance of clearly distinguishing where our conclusions are based on data from the heart and where they are inferred from other systems. We have now made the requested changes, including an amended title, addressed all the criticisms raised, and submitted a thoroughly revised manuscript.

Critique 1:

There are multiple examples where general reviews are cited after descriptions of specific data findings and conclusions. For example, in Section 1.1, it is stated:

"SCs contribute to cardiac repair by supporting the survival and regeneration of cardiac neurons. They are involved in forming protective myelin sheaths around cardiac autonomic neurons, ensuring proper signal transmission that regulates heart rate and rhythm (Hadaya & Ardell, 2020). In response to cardiac injury, SCs can exhibit plasticity, similar to their response in peripheral nerve injury, aiding in the repair and regeneration of damaged cardiac tissue (Jessen & Mirsky, 2019)."

And

"For instance, SC-derived neuregulin-1 has been shown to promote cardiac repair and improve outcomes following myocardial infarction by enhancing cardiomyocyte survival and function (Galindo et al., 2014). This highlights the potential therapeutic applications of targeting SCs in cardiac repair strategies."

And

"Within cardiac ganglia, SGCs form functional networks around neurons and respond to microenvironmental changes, including inflammation or injury, by releasing signalling molecules such as ATP, glutamate, and cytokines. This bidirectional signalling modulates autonomic control of cardiac function,

facilitating physiological adaptation. Disrupted SGC–neuron communication during inflammation has been linked to cardiac dysregulation and arrhythmogenic pathologies (Jessen & Mirsky, 2005; Hanani & Spray, 2020)”

And

“Arrhythmias. SCs ensure proper electrical conduction in the heart. Dysfunction in these cells can lead to impaired myelination and signal transmission, increasing the risk of arrhythmias. By modulating ion homeostasis and neurotransmitter levels, glial cells help maintain the electrophysiological stability of the heart (Jessen & Mirsky, 2005; Guyenet, 2006; Kettenmann & Ransom, 2012)”

The references provided are general reviews. Providing the description from the original papers detailing these findings would strengthen the review.

Response: We thank the reviewer for pointing this out. We have now revised all the statements to cite the original studies.

Critique 2: *The first evidence that identified glial cells in the heart and coined the name term cardiac nexus glia (CNG), also known as cardiac glial cells or cardiac Schwann cells, was published in 2021 (Kikel-Coury et al., 2021).* Please correct/clarify this statement as glial cells were identified in the heart by electron microscopy more than 40 years ago. The original papers should be described and cited.

Response: Thank you for pointing this out. We have now revised this statement.

Critique 3: The conclusion of the heart transplant and reinnervation section (raised as a concern in the first round of review as roles of glia were largely postulated rather than summarized research findings) requires further strengthening. Currently it appears factually stating *“This multifaceted support from glial cells ensures that regenerating neurons can form functional synapses and integrate into existing neural circuits, facilitating the reinnervation of the heart after transplantation (Feng et al., 2012).”* It is not clear how the paper cited is relevant to this statement as the paper is an in vitro paper with no research in heart reinnervation. It would be helpful to clarify the need to provide evidence for synaptic integration and reinnervation facilitation.

Response: We amended the conclusion of this section as follows: “No direct studies have yet examined the role of cardiac glia such as nexus glia, SGCs, or SCs in cardiac reinnervation post-transplantation. However, their known functions in aiding nerve regeneration, establishing functional synapses and integration into existing neural circuits suggest they may influence reinnervation pathways and support regenerating axons. This possibility remains hypothetical and warrants further investigation”. We believe that, even though there is indeed no direct evidence on the role of cardiac glia in reinnervation post-transplantation, keeping this section in the review is essential for precisely that reason – it brings attention to this important research question.

Dear Dr Mastitskaya,

Re: JP-TR-2025-286598R2 "**Glial cells in the heart: implications for their roles in health and disease**" by Svetlana Mastitskaya, Rimma Dugarova, and Shefeeq M Theparambil

We are pleased to tell you that your paper has been accepted for publication in The Journal of Physiology.

Authors should note that it is too late at this point to offer corrections prior to proofing. Major corrections at proof stage, such as changes to figures, will be referred to the Editors for approval before they can be incorporated. Only minor changes, such as to style and consistency, should be made at proof stage. Changes that need to be made after proof stage will usually require a formal correction notice.

Yours sincerely,

Bjorn Knollmann
Senior Editor
The Journal of Physiology

P.S. - You can help your research get the attention it deserves! Check out Wiley's free Promotion Guide for best-practice recommendations for promoting your work at www.wileyauthors.com/eeo/guide. You can learn more about Wiley Editing Services which offers professional video, design, and writing services to create shareable video abstracts, infographics, conference posters, lay summaries, and research news stories for your research at www.wileyauthors.com/eeo/promotion.

IMPORTANT NOTICE ABOUT OPEN ACCESS: To assist authors whose funding agencies mandate public access to published research findings sooner than 12 months after publication, The Journal of Physiology allows authors to pay an Open Access (OA) fee to have their papers made freely available immediately on publication.

You can check if your funder or institution has a Wiley Open Access Account here: <https://authorservices.wiley.com/author-resources/Journal-Authors/licensing-and-open-access/open-access/author-compliance-tool.html>.

EDITOR COMMENTS

Reviewing Editor:

Thank you for the additional revisions to the paper. It is now acceptable for publication.

Senior Editor:

Thank you for your excellent contribution to the Journal!

REFEREE COMMENTS

Referee #2:

Thank you for your efforts in making the requested revisions to this manuscript. These updates have strengthened the content and the scope of the review. I have no further questions or concerns.